# Probabilistic Active Few-Shot Learning in Vision-Language Models

**Anton Baumann**[1]* **Marcus Klasson**[2] **Rui Li**[2] **Arno Solin**[2] **Martin Trapp**[2]

[1]Technical University of Munich [2]Aalto University

## Abstract

Pre-trained vision-language models (VLMs) have shown to be an useful model class for zero- and few-shot learning tasks. In this work, we investigate probabilistic active few-shot learning in VLMs by leveraging post-hoc uncertainty estimation and targeted support set selection. To equip VLMs with a notion of uncertainty on the target task, we utilize a Laplace approximation to the posterior of the VLM and derive a Gaussian approximation to the distribution over the cosine similarities. Further, we propose a simple adaptive target region selection based on k-nearest neighbour search and evaluate on a series of selection strategies from the Bayesian experimental design literature. Our experiments on standard benchmarks show that leveraging epistemic uncertainties leads to improved performance and that further improvements can be obtained by targeting the selection towards the query region.

## 1 Introduction

The rise of foundation models [4, 6, 9, 30] has led to their increasing adoption in downstream tasks where data is scarce [16, 42]. Moreover, in many real-world settings it is imperative that predictions are reliable and that sources of uncertainties are captured and incorporated to avoid failure modes. The paradigm of *active few-shot learning* (or *active fine-tuning*) [1, 17, 40] aims to tackle the challenge of actively selecting a support set (training set for adaption) that is most informative for the downstream task. However, classical approaches, *e.g.*, from the coreset literature [36] or information theory [14], typically do not incorporate all sources of uncertainties into their metric of informativeness. Recent works in Bayesian active learning [15] aim to address this issue by performing selection of support set candidates based on their effect on the epistemic uncertainty of the model [11] or the predictive distribution [3]. Moreover, progress in Bayesian deep learning [29] has resulted in methods that can efficiently estimate epistemic uncertainties in a post-hoc manner [23, 8], making them particularly attractive for active few-shot learning of large scale models.

In this work, we investigate probabilistic active few-shot learning for vision-language models (VLMs) and show benefits of incorporating uncertainties in the support set selection process as well as targeting the selection towards the query region. For this, we propose an uncertainty estimation-based approach by leveraging a Laplace approximation [23] to the posterior of a pre-trained CLIP [30] model. We derive a Gaussian approximation to the distribution over cosine similarities between the image and text embeddings, and investigate different scoring mechanisms for the support set candidate selection. In addition, we propose a simple adaptive target region selection based on $k$-nearest neighbour ($k$-NN) search. In our experiments, we evaluate two few-shot classification settings *(i)* support set selection from a large cross-domain training data source and *(ii)* selection from the training set. We find improved performance over naïve selection for uncertainty-based selection methods and further improvements when the selection is based on an adaptive target region.

---

*Work done during an internship at Aalto University.

Workshop on Bayesian Decision-making and Uncertainty, 38th Conference on Neural Information Processing Systems (NeurIPS 2024).

Fig. 1 illustrates the setting we are considering in this work: Given a pre-trained VLM, we aim to predict labels for a query set of images of a novel downstream task. The VLM agent $\mathcal{M}_0$ is asked to first estimate its uncertainty over the predictions on the query set, where the difficulty of the prediction is proportional to the predictive uncertainty. To avoid failure modes, the agent can select a small number of labelled support set candidates $\mathcal{S}$ from a large data source and use them to update its internal state. Finally, the updated model $\mathcal{M}_1$ is used to predict the labels for the query set.

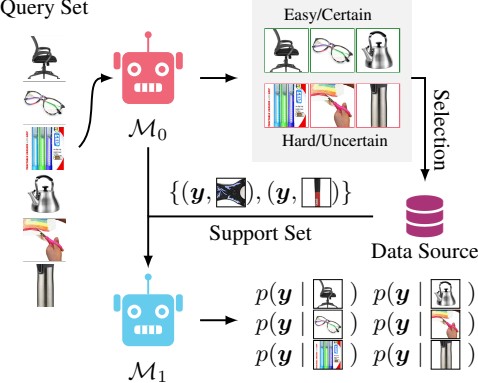

Figure 1: Illustration of the setting.

Our main contributions are the following: *(i)* We propose a post-hoc method for obtaining a distribution over the cosine similarities from a pre-trained VLM without needing architecture changes or further training. *(ii)* We apply our method in active learning and assess various scoring mechanisms for support set selection. *(iii)* We show on benchmark data sets that accounting for epistemic uncertainties improves performance and that targeted candidate selection results in further improvements.

## 2 Methods

We denote vectors by bold lower-case letters (*e.g.*, $\boldsymbol{x}, \boldsymbol{a}$) and use bold upper-case letters for matrices (*e.g.*, $\boldsymbol{X}, \boldsymbol{P}$). Further, sets are denoted in upper-case calligraphic letters (*e.g.*, $\mathcal{D}, \mathcal{I}$) and model parameters or hyper-parameters are denoted using Greek letters (*e.g.*, $\alpha, \boldsymbol{\theta}$). In particular, let $\boldsymbol{x}_i \in \mathbb{R}^{p_{\text{IMG}}}$ and $\boldsymbol{y}_j \in \mathbb{R}^{p_{\text{TXT}}}$ denote the $i^{\text{th}}$ image and $j^{\text{th}}$ text description, respectively. Further, we use $\phi : \mathbb{R}^{p_{\text{IMG}}} \to \mathbb{R}^{d_{\text{IMG}}}$ and $\psi : \mathbb{R}^{p_{\text{TXT}}} \to \mathbb{R}^{d_{\text{TXT}}}$ to denote the image and text encoders of the VLM, where $p_{\text{IMG}}$ and $p_{\text{TXT}}$ denote the respective input dimensionality and $d_{\text{IMG}}, d_{\text{TXT}}$ is the dimensionality of the respective feature space. The embeddings are projected into a joint space, given as $\boldsymbol{g} = \boldsymbol{P}\phi(\boldsymbol{x})$ and $\boldsymbol{h} = \boldsymbol{Q}\psi(\boldsymbol{y})$, using linear projections denoted by $\boldsymbol{P} \in \mathbb{R}^{d \times d_{\text{IMG}}}$ and $\boldsymbol{Q} \in \mathbb{R}^{d \times d_{\text{TXT}}}$, respectively.

VLMs (*e.g.*, [30]) are typically trained by minimizing the InfoNCE loss [28], which is the sum of two cross-entropy terms, one for each relational direction—image to text (IMG $\to$ TXT) or text to image (IMG $\leftarrow$ TXT). The loss is given as $\mathcal{L}(\boldsymbol{X}, \boldsymbol{Y}) = \frac{1}{2}\mathcal{L}_{\text{CE}}^{\text{IMG} \to \text{TXT}}(\boldsymbol{X}, \boldsymbol{Y}) + \frac{1}{2}\mathcal{L}_{\text{CE}}^{\text{IMG} \leftarrow \text{TXT}}(\boldsymbol{X}, \boldsymbol{Y})$ with cross-entropy loss terms defined over the cosine similarities between the embeddings, *i.e.*,

$$\mathcal{L}_{\text{CE}}^{\text{IMG} \to \text{TXT}}(\boldsymbol{X}, \boldsymbol{Y}) = \sum_{i=1}^{n} -\log\left(\frac{\exp(\hat{\boldsymbol{g}}_i^\top \hat{\boldsymbol{h}}_i)}{\sum_{j=1}^{n} \exp(\hat{\boldsymbol{g}}_i^\top \hat{\boldsymbol{h}}_j)}\right), \tag{1}$$

where $\hat{\boldsymbol{g}}$ and $\hat{\boldsymbol{h}}$ are the unit-length normalized embeddings. For further details see App. B.1.

In this work, we utilize post-hoc uncertainty estimation based on the Laplace approximation [23] to estimate uncertainties over the model parameters. This approach has found increasing application in contemporary deep learning (*e.g.*, [8, 20, 25]) and uses a Gaussian approximation to the posterior distribution. Utilising a Laplace approximation allows us to induce uncertainty over the feature embeddings of both encoders and results in a distribution over cosine similarities, which in turn enables quantifying model uncertainties in a principled manner. Fig. 2 illustrates the propagation of uncertainties in our setup by estimating uncertainties over the projection matrices.

**Laplace approximation** One of the main computational challenges associated with the Laplace approximation is related to the estimation of the Hessian matrix of the log joint w.r.t. the model parameters. Since a naïve approach is computationally impractical in the case of VLMs, we chose to estimate the Kronecker-factored Generalized Gauss–Newton (GGN) approximation [33, 24]. Moreover, we apply the Laplace approximation only for the projection matrices $\boldsymbol{P}$ and $\boldsymbol{Q}$ of the image and text encoders. Hence, resulting in GGN approximations $\text{GGN}_{\text{IMG}}$ and $\text{GGN}_{\text{TXT}}$ given in form of their Kronecker factors, see App. C.1 for details.

However, naïvely applying Laplace approximations in VLMs is challenging as the contrastive loss entangles $\boldsymbol{P}$ and $\boldsymbol{Q}$, which further complicates the estimation of the Hessian. These models are

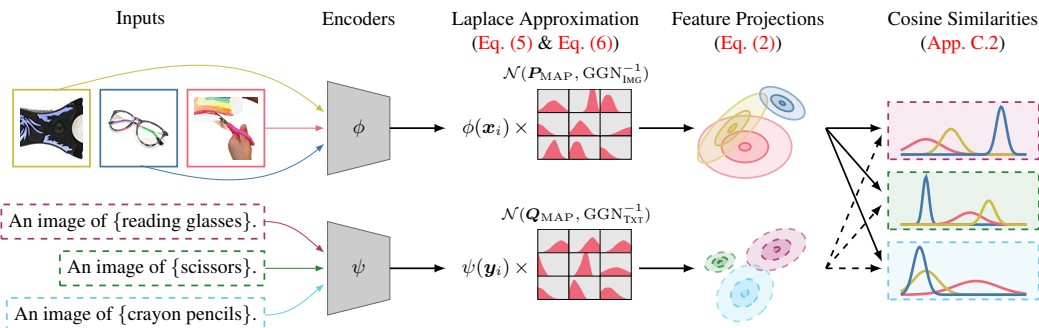

Figure 2: Illustration of uncertainty propagation in VLMs: We estimate uncertainties over the projection matrices of both encoders using a Laplace approximation, which induces distributions over the feature projections. We then approximate the distribution over cosine similarities by a Gaussian.

also typically trained with mini-batch sizes of around $30k$ samples. In order to compute the GGN approximations in VLMs, we simplify the contrastive loss $\mathcal{L}$ used for pre-training by assuming independence between $\boldsymbol{P}$ and $\boldsymbol{Q}$. Specifically, we treat each of the two loss terms independent and consider only $\mathcal{L}_{\text{CE}}^{\text{IMG}\to\text{TXT}}$ for the image encoder and $\mathcal{L}_{\text{CE}}^{\text{IMG}\leftarrow\text{TXT}}$ for the text encoder in the Laplace approximation. Hence, dropping interactions between the image and text encoders in the Laplace approximation. Lastly, we use an incremental computation of the Kronecker factors to account for large mini-batch sizes. Further details and derivations are given in App. C.1.

**Distribution over cosine similarities** As the Laplace approximation uses a Gaussian approximation, the feature embeddings are distributed according to another Gaussian distribution. Specifically, the distribution over embedding vectors $\boldsymbol{g}$ (or $\boldsymbol{h}$) for a datum $\boldsymbol{x}$ (or $\boldsymbol{y}$) can be expressed as follows due to linearity, *i.e.*,

$$\mathcal{N}\left(\boldsymbol{g}, \left(\phi(\boldsymbol{x})^\top \boldsymbol{A}_{\text{IMG}}^{-1}\phi(\boldsymbol{x})\right)\boldsymbol{B}_{\text{IMG}}^{-1}\right) \quad \text{and} \quad \mathcal{N}\left(\boldsymbol{h}, \left(\psi(\boldsymbol{y})^\top \boldsymbol{A}_{\text{TXT}}^{-1}\psi(\boldsymbol{y})\right)\boldsymbol{B}_{\text{TXT}}^{-1}\right), \quad (2)$$

where $\boldsymbol{A}$ and $\boldsymbol{B}$ denote the Kronecker factors of the GNN approximation of the Hessian matrix, respectively. Unfortunately, the distribution over cosine similarities is in general not Gaussian. However, by assuming independence between the elements of $\boldsymbol{g}$ and $\boldsymbol{h}$ and in the limit of $d \to \infty$ we can approximate the distribution over cosine similarities to be Gaussian distributed. We find this approximation to work well in practice, while not accurately capturing the skewness of the distributions. A detailed derivation and empirical results on the approximation quality are given in App. C.2.

**Targeted support set selection** Let $\mathcal{X}_{\text{test}} = \{\boldsymbol{x}_i^*\}$ with $\boldsymbol{x}_i^* \sim p(\boldsymbol{x}^*)$ be a set of unseen test data (query set) with unknown class labels. We aim to find a set $\{(\boldsymbol{x}_j, \boldsymbol{y}_j)\}_j^m$ of support candidates of cardinality $m$ with $\boldsymbol{x}_j, \boldsymbol{y}_j \sim p(\boldsymbol{x}_j, \boldsymbol{y}_j)$ such that we reduce uncertainty over the class labels of $\mathcal{X}_{\text{test}}$. To approach this problem, we target the selection process towards the predictive distribution of the query set. In particular, we propose to use a $k$-nearest neighbours selection in the joint space to pre-select support set candidates based on the Wasserstein distance between the distributions over image embeddings. After pre-selection, we quantify the information gain of the support set candidates either using the entropy over the predictive distribution, the expected predictive information gain (EPIG, [3]), or the BALD score [15]. Doing so adaptively targets the candidate search for the the support set towards the predictive distribution of the query set and reduces the computational complexity of the selection process. Further details on the selection process and the score functions are given in App. D.

## 3 Experiments

To evaluate our approach for probabilistic active few-show learning, we conducted experiments using pre-trained OpenCLIP models from Hugging Face [18]. We estimated the Laplace approximations of the OpenCLIP model with ViT-Base backbone and ViT-Huge backbone [10] using a randomly sampled subset from the Laion-400M data set [35]. Further details are given in App. E.

For probabilistic active few-shot learning with VLMs we consider the task of image classification and present results on the Flowers102 [27], Food101 [5], CIFAR-100 [21], ImageNet-R [13], EuroSAT

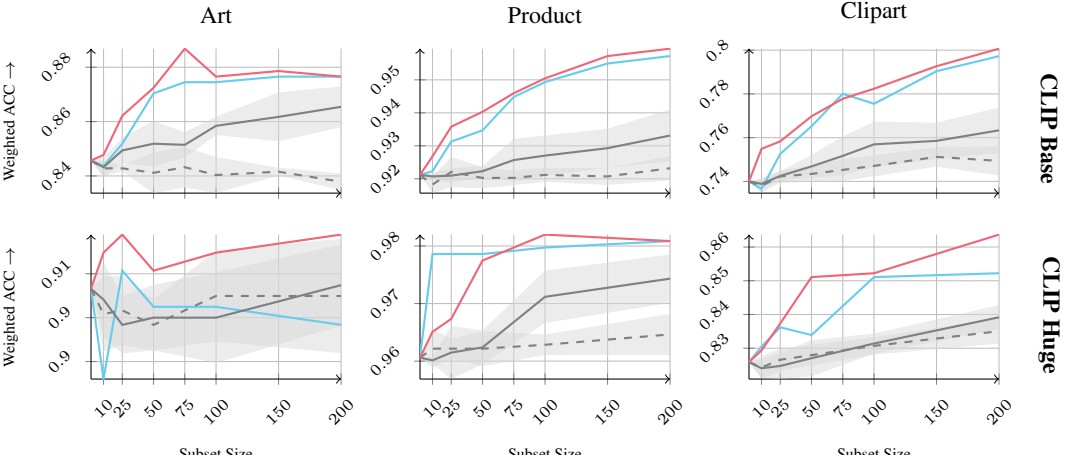

Figure 3: Results on the Office-Home data set with support set selection from all training domains. We observe that incorporating epistemic uncertainties (——) improves over entropy based targeted selection (——) in most of the cases and outperforms naïve random selection (- - -) and random selection with targeted support set candidates (——). Shaded regions indicate the std over 5 runs.

[12], and the Office-Home [39] data sets. To assess the performance of the proposal, we investigated the following questions: *(i)* Do approaches that account for epistemic uncertainties improve performance? *(ii)* What is the effect of targeting the support set candidates towards the query region? *(iii)* How does the model capacity affect the performance of the proposed approach?

To address these questions, we performed support set selection from all training domains available in the Office-Home data set and evaluated on the test set (query set) of each domain independently. In Fig. 3 we compare the performance of targeted entropy-based support set selection, random selection, random selection with targeted support set region, and the best performing (according to the validation loss) acquisition function that incorporates epistemic uncertainties. We find that incorporating epistemic uncertainties improves the few-shot learning performance in most cases and generally outperforms random selection. Further, we observe that targeted support set selection improves the performance as indicated by the performance gap between naïve random selection and targeted random selection and that the model capacity can have a substantial impact on the performance gains across all approaches. A listing of the results using the negative log-predictive density are given in App. E.2.

**Single-domain Finetuning**   In App. E.2, we show results for single-domain finetuning on standard benchmark data sets (*e.g.* CIFAR-100, Imagenet-R, Flowers102, etc.) using the different support set selection methods with the OpenCLIP model. The selection methods using the epistemic uncertainty (BALD and EPIG) perform better or on par with the Targeted Maximum Entropy across the different subset sizes and data sets, which demonstrates the benefits of using our proposed uncertainty estimates for support set selection.

## 4   Discussion and Conclusion

In this work, we have introduced a probabilistic active few-shot learning approach for VLMs. Our approach leverages a Laplace approximation to the posterior of the projection layers of the VLM to estimate epistemic uncertainties. We have further introduced an adaptive targeted support set candidate selection based on $k$-NN selection using the Wasserstein distance between the distributions over image embeddings in the joint space. To assess the performance of probabilistic active few-shot learning in VLMs, we have conducted two sets of experiments, one in the cross-domain setting on the Office-Home data set and one in the single-domain setting on standard benchmark data sets. We found that incorporating epistemic uncertainties improves the few-shot learning performance in most cases and generally outperforms random selection. Moreover, targeting the selection process towards the query region provides further improvements in all cases.

**Reproducibility** The code for the experiments is available at: https://aaltoml.github.io/BayesVLM/.

**Acknowledgements**

AS and RL acknowledge funding from the Research Council of Finland (grant number 339730). MT acknowledges funding from the Research Council of Finland (grant number 347279). MK acknowledge funding from the Finnish Center for Artificial Intelligence (FCAI). We acknowledge CSC – IT Center for Science, Finland, for awarding this project access to the LUMI supercomputer, owned by the EuroHPC Joint Undertaking, hosted by CSC (Finland) and the LUMI consortium through CSC. We acknowledge the computational resources provided by the Aalto Science-IT project.

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

# Probabilistic Active Few-Shot Learning in Vision-Language Models

## Supplementary Material

## A  Related Work

### A.1  Active Learning

The active learning setting [32] entails an agent learning a task from an unlabelled dataset, while simultaneously determining which data points to label for maximal benefit to the target task. The learner uses an acquisition function to base its sample selection on that should quantify how beneficial (or informative) this sample will be to learn from for the target task. There exist various acquisition functions, *e.g.*, (i) entropy-based which aims to minimize the expected entropy after observing data points [14], and (ii) core-set based methods which are trained to minimize the generalization error between the unlabelled and labelled sets and use clustering for selection [36]. Uncertainty-based acquisition functions have been explored to select data points that will mostly reduce the epistemic uncertainty in the model, *e.g.*, Bayesian Active Learning by Disagreement (BALD) score [11, 15]. More recently, the expected predictive information gain (EPIG) [3] was proposed to measure the information gain in the space of predictions rather than parameters. We experiment with the mentioned uncertainty-based acquisition functions combined with our probabilistic embeddings for targeted data selection in VLM finetuning.

### A.2  Probabilistic Vision-Language Models

Several works are aiming to extend VLMs to produce predictive uncertainty estimates for various downstream tasks, *e.g.*, cross-modal retrieval [7, 22] and visual-question answering [19]. These methods learn probabilistic embeddings on each modality by estimating probability distributions from the network. However, this approach requires training the networks from scratch, which limits their applicability to pretrained VLMs ( *e.g.* CLIP). To this end, Upadhyay et al. [38] proposed a post-hoc method called ProbVLM that learns probabilistic embeddings from finetuned adapters on a frozen VLM backbone. Similar to this work, they also apply their method to the active learning task and use the uncertainty estimates for selecting informative subsets of training data for finetuning. However, ProbVLM requires finetuning the probabilistic embeddings on a proxy task, while our method can be applied directly on the pretrained model.

## B  Preliminaries

This section provides a brief overview of the background concepts relevant to this work.

### B.1  Vision-Language Models

In this work, we consider vision-language models (VLM) learned using the contrastive learning objective known as InfoNCE. In particular, let $x_i \in \mathbb{R}^{p_{\text{IMG}}}$ and $y_j \in \mathbb{R}^{p_{\text{TXT}}}$ denote the $i$th image and $j$th text description, respectively. Further, we use $\phi : \mathbb{R}^{p_{\text{IMG}}} \to \mathbb{R}^{d_{\text{IMG}}}$ and $\psi : \mathbb{R}^{p_{\text{TXT}}} \to \mathbb{R}^{d_{\text{TXT}}}$ to denote the image and text encoders of the VLM, where $p_{\text{IMG}}$ and $p_{\text{TXT}}$ denote the respective input dimensionalities and $d_{\text{IMG}}$, $d_{\text{TXT}}$ is the dimensionality of the respective feature space.

To project the embeddings into a joint space, we assume a linear projection layer for both the image and the text encoder denoted by $\boldsymbol{P} \in \mathbb{R}^{d \times d_{\text{IMG}}}$ and $\boldsymbol{Q} \in \mathbb{R}^{d \times d_{\text{TXT}}}$, respectively. The embeddings in the joint space are then given as $\boldsymbol{g}_i = \boldsymbol{P}\phi(\boldsymbol{x}_i)$ and $\boldsymbol{h}_j = \boldsymbol{Q}\psi(\boldsymbol{y}_j)$ and we use hat notation to denote the unit-length normalized embeddings, e.g., $\hat{\boldsymbol{g}}_i = \frac{\boldsymbol{P}\phi(\boldsymbol{x}_i)}{\|\boldsymbol{P}\phi(\boldsymbol{x}_i)\|}$.

VLM models ( e.g., [30]) are typically trained by minimizing the InfoNCE loss, which is given as the sum of two cross-entropy terms, one for each relational direction – image to text (IMG $\rightarrow$ TXT) or text to image (IMG $\leftarrow$ TXT). Specifically, the InfoNCE loss is given as $\mathcal{L}(\boldsymbol{X}, \boldsymbol{Y}) = \frac{1}{2}\mathcal{L}_{\text{CE}}^{\text{IMG} \rightarrow \text{TXT}}(\boldsymbol{X}, \boldsymbol{Y}) + \frac{1}{2}\mathcal{L}_{\text{CE}}^{\text{IMG} \leftarrow \text{TXT}}(\boldsymbol{X}, \boldsymbol{Y})$ with cross-entropy loss terms given as:

$$\mathcal{L}_{\text{CE}}^{\text{IMG} \rightarrow \text{TXT}}(\boldsymbol{X}, \boldsymbol{Y}) = \sum_{i=1}^{n} -\log \left( \frac{\exp(\hat{\boldsymbol{g}}_i^\top \hat{\boldsymbol{h}}_i)}{\sum_{j=1}^{n} \exp(\hat{\boldsymbol{g}}_i^\top \hat{\boldsymbol{h}}_j)} \right) \qquad (3)$$

$$\mathcal{L}_{\text{CE}}^{\text{IMG} \leftarrow \text{TXT}}(\boldsymbol{X}, \boldsymbol{Y}) = \sum_{i=1}^{n} -\log \left( \frac{\exp(\hat{\boldsymbol{h}}_i^\top \hat{\boldsymbol{g}}_i)}{\sum_{j=1}^{n} \exp(\hat{\boldsymbol{h}}_i^\top \hat{\boldsymbol{g}}_j)} \right). \qquad (4)$$

For further details we refer the reader to [30, 41]

## B.2 Bayesian Deep Learning

We will briefly review concepts on Bayesian deep learning relevant to this work. Given a dataset $\mathcal{D} = \{(\boldsymbol{x}_i, \boldsymbol{y}_i)\}_{i=1}^{n}$ and a probabilistic models with likelihood function $p(\boldsymbol{y} \mid \boldsymbol{x}, \boldsymbol{\theta})$ and prior distribution $p(\boldsymbol{\theta})$, we aim to estimate the posterior distribution $p(\boldsymbol{\theta} \mid \mathcal{D})$ of the model parameters $\boldsymbol{\theta}$ given the training data $\mathcal{D}$. In the context of deep learning, exact inference of the posterior distribution is at least NP-hard in most settings and only becomes tractable if $p(\boldsymbol{\theta} \mid \mathcal{D})$ constitute sufficient structure [23]. Henceforth, we consider approximate Bayesian inference using the Laplace approximation [23] in this work, which has gained increasing popularity in the community ( e.g., [33, 8, 25, 34]) as a post-hoc techniques to estimate epistemic uncertainties.

The Laplace approximation uses a second-order Taylor expansion of the log-joint around the maximum-a-posteriori (MAP) estimate $\boldsymbol{\theta}_{\text{MAP}}$. The resulting distribution is then approximated with an un-normalised Gaussian density, which in turn results in an approximate posterior distribution given by a Gaussian distribution located at the MAP estimate, i.e., $p(\boldsymbol{\theta} \mid \mathcal{D}) \approx \mathcal{N}(\boldsymbol{\theta} \mid \boldsymbol{\theta}_{\text{MAP}}, \boldsymbol{\Sigma})$. Resulting from the Taylor expansion, the covariance is given by the inverse Hessian at the MAP, i.e., $\boldsymbol{\Sigma} = (-\nabla^2 \log p(\boldsymbol{\theta}, \mathcal{D})|_{\boldsymbol{\theta}=\boldsymbol{\theta}_{\text{MAP}}})^{-1}$. Predictions are then made based on the posterior predictive distribution $p(\boldsymbol{y} \mid \boldsymbol{x}, \mathcal{D}) = \int p(\boldsymbol{y} \mid \boldsymbol{x}, \boldsymbol{\theta}) p(\boldsymbol{\theta} \mid \mathcal{D}) \mathrm{d}\boldsymbol{\theta}$, which is typically performed by Monte Carlo sampling in case of non-linear likelihoods functions, e.g., classification settings. We refer to [8] for a detailed review of the topic.

# C Derivations

This section provides detailed derivations of the equations presented in the main text.

## C.1 Laplace Approximation

To obtain the Laplace approximation to the VLM, we first assume independence between $\boldsymbol{P}$ and $\boldsymbol{Q}$. The resulting GGN approximations $\text{GGN}_{\text{IMG}}$ and $\text{GGN}_{\text{TXT}}$ are then given in form of their Kronecker factors $\boldsymbol{A}$ and $\boldsymbol{B}$, i.e.,

$$\text{GGN}_{\text{IMG}} \approx \underbrace{\left[ \frac{1}{\sqrt{n}} \sum_{i=1}^{n} \phi(\boldsymbol{x}_i)\phi(\boldsymbol{x}_i)^\top \right]}_{=\boldsymbol{A}_{\text{IMG}}} \otimes \underbrace{\left[ \frac{1}{\sqrt{n}} \sum_{i=1}^{n} J_{\text{IMG}}(\boldsymbol{x}_i)^\top \boldsymbol{\Lambda}_{\text{IMG}} J_{\text{IMG}}(\boldsymbol{x}_i) \right]}_{=\boldsymbol{B}_{\text{IMG}}}, \qquad (5)$$

where $J_{\text{IMG}}(\boldsymbol{x}_i) = \frac{\partial \hat{\boldsymbol{g}}_i^\top \hat{\boldsymbol{H}}}{\partial \boldsymbol{g}_i}$ and

$$\text{GGN}_{\text{TXT}} \approx \underbrace{\left[ \frac{1}{\sqrt{n}} \sum_{i=1}^{n} \psi(\boldsymbol{y}_i)\psi(\boldsymbol{y}_i)^\top \right]}_{=\boldsymbol{A}_{\text{TXT}}} \otimes \underbrace{\left[ \frac{1}{\sqrt{n}} \sum_{i=1}^{n} J_{\text{TXT}}(\boldsymbol{x}_i)^\top \boldsymbol{\Lambda}_{\text{TXT}} J_{\text{TXT}}(\boldsymbol{x}_i) \right]}_{=\boldsymbol{B}_{\text{TXT}}}, \qquad (6)$$

where $n$ is the number image-text pairs in the training set.

We further incorporate the prior precision $\lambda$ into the GGN approximation by adding the prior precision to the diagonal of the GGN Hessian, *i.e.*,

$$\text{GGN}_{\text{IMG}} \approx \tau \left( \boldsymbol{A}_{\text{IMG}} \otimes \boldsymbol{B}_{\text{IMG}} \right) + \lambda \boldsymbol{I} \tag{7}$$

$$\approx \left( \sqrt{\tau} \boldsymbol{A}_{\text{IMG}} + \sqrt{\lambda} \boldsymbol{I} \right) \otimes \left( \sqrt{\tau} \boldsymbol{B}_{\text{IMG}} + \sqrt{\lambda} \boldsymbol{I} \right). \tag{8}$$

In our experiments, we set $\tau = 0.75$ for the ViT-Base model and $\tau = 0.3$ for the ViT-Huge model and obtain the prior precision $\lambda$ through marginal likelihood maximization.

### C.1.1 Obtaining the Posterior Predictive Distribution

For conciseness, we denote the posterior precision matrices associated with the image encoder as $\boldsymbol{A}_{\text{IMG}}$ and $\boldsymbol{B}_{\text{IMG}}$. We have obtained the posterior distribution over the image projection matrix $\boldsymbol{P}$ represented as $\mathcal{N}(\text{vec}(\boldsymbol{P}); \text{vec}(\boldsymbol{P}_{\text{MAP}}), \text{GGN}_{\text{IMG}}^{-1})$. Given that $\text{GGN}_{\text{IMG}}^{-1}$ is formulated using the Kronecker product of the inverses of these matrices, *i.e.*, $\boldsymbol{A}_{\text{IMG}}^{-1} \otimes \boldsymbol{B}_{\text{IMG}}^{-1}$, we proceed to express the posterior predictive distribution as a matrix normal distribution $\mathcal{MN}(\boldsymbol{P}; \boldsymbol{P}_{\text{MAP}}, \boldsymbol{B}_{\text{IMG}}^{-1}, \boldsymbol{A}_{\text{IMG}}^{-1})$ as referenced in [2]:

$$\boldsymbol{P} \sim \mathcal{MN}(\boldsymbol{P}_{\text{MAP}}, \boldsymbol{B}_{\text{IMG}}^{-1}, \boldsymbol{A}_{\text{IMG}}^{-1}) \tag{9}$$

$$\implies \boldsymbol{g} = \boldsymbol{P}\phi(\boldsymbol{x}) \sim \mathcal{MN}(\boldsymbol{P}_{\text{MAP}}\phi(\boldsymbol{x}), \boldsymbol{B}_{\text{IMG}}^{-1}, \phi(\boldsymbol{x})^\top \boldsymbol{A}_{\text{IMG}}^{-1} \phi(\boldsymbol{x})) \tag{10}$$

$$\implies \boldsymbol{g} \sim \mathcal{N}(\boldsymbol{P}_{\text{MAP}}\boldsymbol{a}, \left( \phi(\boldsymbol{x})^\top \boldsymbol{A}_{\text{IMG}}^{-1} \phi(\boldsymbol{x}) \right) \boldsymbol{B}_{\text{IMG}}^{-1}) \tag{11}$$

### C.1.2 Online Laplace Approximation

For the EPIG score, we update our Laplace approximation online after each data point is added to the support set. Given the current Laplace approximation of the posterior over the image projection matrix $\boldsymbol{P}$ we update the posterior distribution as follows:

$$\boldsymbol{P}_{t+1} = \boldsymbol{P}_t - \gamma \nabla_{\boldsymbol{P}} \mathcal{L}_{\text{CE}}^{\text{IMG} \rightarrow \text{TXT}}(\boldsymbol{x}^*, \boldsymbol{Y}) \tag{12}$$

$$\boldsymbol{A}_{\text{IMG},t+1} = \boldsymbol{A}_{\text{IMG},t} + \beta \phi(\boldsymbol{x}^*)\phi(\boldsymbol{x}^*)^\top \tag{13}$$

$$\boldsymbol{B}_{\text{IMG},t+1} = \boldsymbol{B}_{\text{IMG},t} + \beta J_{\text{IMG}}(\boldsymbol{x}^*)^\top \boldsymbol{\Lambda}_{\text{IMG}} J_{\text{IMG}}(\boldsymbol{x}^*) \tag{14}$$

From the updated $\boldsymbol{A}_{\text{IMG},t+1}$ and $\boldsymbol{B}_{\text{IMG},t+1}$ we obtain the updated GGN approximation of the Hessian matrix:

$$\text{GGN}_{\text{IMG},t+1} \approx \left( \sqrt{\tau} \boldsymbol{A}_{\text{IMG},t+1} + \sqrt{\lambda} \boldsymbol{I} \right) \otimes \left( \sqrt{\tau} \boldsymbol{B}_{\text{IMG},t+1} + \sqrt{\lambda} \boldsymbol{I} \right) \tag{15}$$

After each update, we optimize for the prior precision $\lambda$ by maximizing the marginal likelihood. For our experiments, we set the learning rates $\gamma = 10^{-3}$ and $\beta = 1$.

### C.1.3 Jacobians for the GGN Approximation

In the following we derive the Jacobians $J_{\text{IMG}}(\boldsymbol{x}_i)$ and $J_{\text{TXT}}(\boldsymbol{y}_i)$ used in the Kronecker-factored Generalized Gauss-Newton (GGN) approximation of the Hessian matrices. Let $\hat{\boldsymbol{g}}_i$ and $\hat{\boldsymbol{h}}_j$ denote the normalized image and text embedding, respectively. With some misuse of notation, let $\hat{\boldsymbol{H}}$ denote the matrix of normalized text embeddings with $\hat{\boldsymbol{h}}_j$ as its columns, and $\hat{\boldsymbol{G}}$ the matrix of normalized image embeddings with $\hat{\boldsymbol{g}}_i$ as its columns. Then, for the InfoNCE likelihood, which depends on the dot product between the normalized embedding in the batch, we compute the Jacobian for the image encoder as follows:

$$J_{\text{IMG}}(\boldsymbol{x}_i)^\top = \frac{\partial \hat{\boldsymbol{H}}^\top \hat{\boldsymbol{g}}_i}{\partial \boldsymbol{g}_i} = \hat{\boldsymbol{H}}^\top \frac{\partial}{\partial \boldsymbol{g}_i} \frac{\boldsymbol{g}_i}{\|\boldsymbol{g}_i\|} = \hat{\boldsymbol{H}}^\top \frac{\|\boldsymbol{g}_i\| - \boldsymbol{g}_i \frac{\partial \|\boldsymbol{g}_i\|}{\partial \boldsymbol{g}_i}}{\|\boldsymbol{g}_i\|^2} = \hat{\boldsymbol{H}}^\top \frac{\|\boldsymbol{g}_i\| - \frac{\boldsymbol{g}_i \boldsymbol{g}_i^\top}{\|\boldsymbol{g}_i\|}}{\|\boldsymbol{g}_i\|^2} \tag{16}$$

$$= \hat{\boldsymbol{H}}^\top \left( \frac{1}{\|\boldsymbol{g}_i\|} - \frac{\boldsymbol{g}_i \boldsymbol{g}_i^\top}{\|\boldsymbol{g}_i\|^3} \right) \tag{17}$$

Analogously, we obtain the Jacobian for the text encoder as:

$$J_{\text{TXT}}(\boldsymbol{y}_i)^\top = \hat{\boldsymbol{G}}^\top \left( \frac{1}{\|\boldsymbol{h}_i\|} - \frac{\boldsymbol{h}_i \boldsymbol{h}_i^\top}{\|\boldsymbol{h}_i\|^3} \right) \tag{18}$$

### C.1.4 Likelihood Hessian for the GGN Approximation

The zero-shot classifier induced by CLIP computes unnormalized logits for each class $c$, represented by $\hat{\boldsymbol{g}}_i^\top \hat{\boldsymbol{h}}_c =: f_c$. By applying the softmax function, we calculate the probabilities for each class $c$ as $\pi_c = \frac{\exp(f_c)}{\sum_{c'} \exp(f_{c'})}$. The likelihood Hessian of the cross-entropy loss for this classifier is represented by:

$$\Lambda_{\mathrm{IMG}} = \mathrm{diag}(\boldsymbol{\pi}) - \boldsymbol{\pi}\boldsymbol{\pi}^\top \tag{19}$$

Similarly, the likelihood Hessian for the text encoder follows analogous principles in the text-to-image direction. For a more detailed derivation of the likelihood Hessian, we refer to [31]. Rearranging terms in the analytical expression for $J_{\mathrm{IMG}}^\top \Lambda_{\mathrm{IMG}} J_{\mathrm{IMG}}$ facilitates space-efficient computation of the GGN approximation.

## C.2 Distribution over Cosine Similarities

For the derivation of the distribution over cosine similarities, first recall the definition of the cosine similarity between two vectors, $\boldsymbol{g}$ and $\boldsymbol{h}$, which is given as $\mathrm{S}_{\cos}(\boldsymbol{g}, \boldsymbol{h}) = \frac{\boldsymbol{g}^\top \boldsymbol{h}}{\|\boldsymbol{g}\|\|\boldsymbol{h}\|}$. Now, with some abuse of notation, let $\boldsymbol{g}$ and $\boldsymbol{h}$ denote random vectors for the image and text embeddings, respectively. Further, let us assume that their distribution follows a Gaussian distribution with mean $\boldsymbol{\mu_g} = (\mu_{\boldsymbol{g},1}, \ldots, \mu_{\boldsymbol{g},d})$ and $\boldsymbol{\mu_h} = (\mu_{\boldsymbol{h},1}, \ldots, \mu_{\boldsymbol{h},d})$ and diagonal covariance structure, *i.e.*, $\boldsymbol{\Sigma_g} = \mathrm{diag}(\sigma_{\boldsymbol{g},1}^2, \ldots, \sigma_{\boldsymbol{g},d}^2)$ and $\boldsymbol{\Sigma_h} = \mathrm{diag}(\sigma_{\boldsymbol{h},1}^2, \ldots, \sigma_{\boldsymbol{h},d}^2)$.

Then the expected value of the cosine similarity is:

$$\mathbb{E}[\mathrm{S}_{\cos}(\boldsymbol{g}, \boldsymbol{h})] = \frac{\mathbb{E}[\boldsymbol{g}^\top \boldsymbol{h}]}{\mathbb{E}[\|\boldsymbol{g}\|]\mathbb{E}[\|\boldsymbol{h}\|]} \tag{20}$$

$$= \frac{\sum_i^d \mu_{\boldsymbol{g},i}\mu_{\boldsymbol{h},i}}{\mathbb{E}[\|\boldsymbol{g}\|]\mathbb{E}[\|\boldsymbol{h}\|]}. \tag{21}$$

Note that computing $\mathbb{E}[\|\boldsymbol{x}\|]$ is intractable, and we therefore bound the expected value by application of the triangle inequality, *i.e.*,

$$\mathbb{E}[\|\boldsymbol{x}\|] \leq \sqrt{\sum_i \mu_{\boldsymbol{x},i}^2 + \sigma_{\boldsymbol{x},i}^2}, \tag{22}$$

where we use the fact that $\mathbb{E}[x^2] = \mu_x^2 + \sigma_x^2$. Consequently, we obtain an approximation to the expected value of the cosine similarity given by:

$$\mathbb{E}[\mathrm{S}_{\cos}(\boldsymbol{g}, \boldsymbol{h})] \approx \frac{\sum_i^d \mu_{\boldsymbol{g},i}\mu_{\boldsymbol{h},i}}{\sqrt{\sum_i \mu_{\boldsymbol{g},i}^2 + \sigma_{\boldsymbol{g},i}^2}\sqrt{\sum_i \mu_{\boldsymbol{h},i}^2 + \sigma_{\boldsymbol{h},i}^2}}. \tag{23}$$

Next, we will derive the second moment (variance) of the cosine similarity of two random vectors. First note that the variance can be written as the difference of two expectations, *i.e.*,

$$\mathbb{V}\mathrm{ar}[\mathrm{S}_{\cos}(\boldsymbol{g}, \boldsymbol{h})] = \mathbb{E}[\mathrm{S}_{\cos}(\boldsymbol{g}, \boldsymbol{h})^2] - \mathbb{E}[\mathrm{S}_{\cos}(\boldsymbol{g}, \boldsymbol{h})]^2, \tag{24}$$

where the second expection corresponds to:

$$\mathbb{E}[\mathrm{S}_{\cos}(\boldsymbol{g}, \boldsymbol{h})]^2 \approx \frac{(\sum_i^d \mu_{\boldsymbol{g},i}\mu_{\boldsymbol{h},i})^2}{\sum_i \mu_{\boldsymbol{g},i}^2 + \sigma_{\boldsymbol{g},i}^2 \sum_i \mu_{\boldsymbol{h},i}^2 + \sigma_{\boldsymbol{h},i}^2}. \tag{25}$$

Next we can obtain $\mathbb{E}[\mathrm{S}_{\cos}(\boldsymbol{g}, \boldsymbol{h})^2]$ for which we will use the fact that $\mathbb{E}[x^2] = \mu_x^2 + \sigma_x^2$ again, *i.e.*,

$$\mathbb{E}[\mathrm{S}_{\cos}(\boldsymbol{g}, \boldsymbol{h})^2] = \frac{\mathbb{E}[(\boldsymbol{g}^\top \boldsymbol{h})^2]}{\sum_i \mu_{\boldsymbol{g},i}^2 + \sigma_{\boldsymbol{g},i}^2 \sum_i \mu_{\boldsymbol{h},i}^2 + \sigma_{\boldsymbol{h},i}^2} \tag{26}$$

where

$$\mathbb{E}[(\boldsymbol{g}^\top \boldsymbol{h})^2] = \sum_i \sum_j \mu_{\boldsymbol{g},i}\mu_{\boldsymbol{h},i}\mu_{\boldsymbol{g},j}\mu_{\boldsymbol{h},j} \tag{27}$$

$$+ \sum_i \sigma_{\boldsymbol{g},i}^2\mu_{\boldsymbol{h},i}^2 + \mu_{\boldsymbol{g},i}^2\sigma_{\boldsymbol{h},i}^2 + \sigma_{\boldsymbol{g},i}^2\sigma_{\boldsymbol{h},i}^2. \tag{28}$$

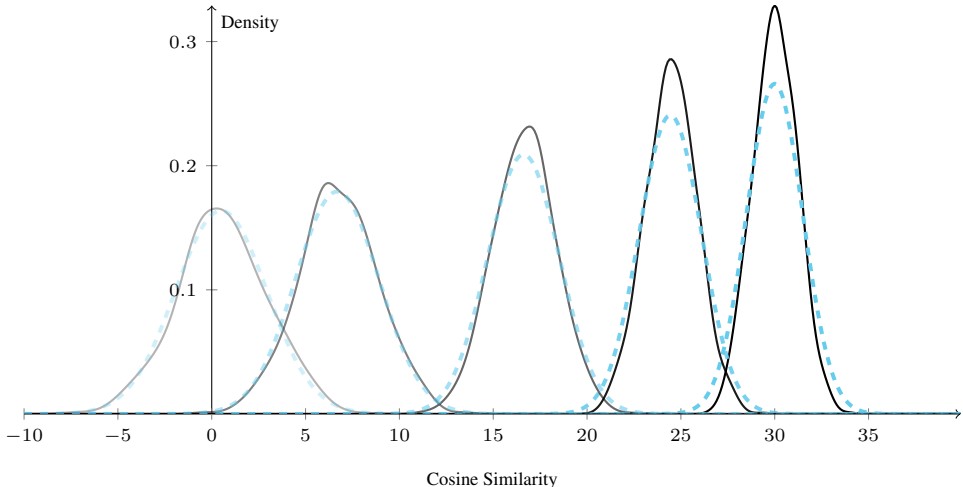

Figure 4: Approximation quality of the Gaussian approximation (- - -) to the distribution over cosine similarities compared to KDE over samples (——) for image-text pairs with increasing Euclidean distance between their feature projection means $(\boldsymbol{\mu_g}, \boldsymbol{\mu_h})$.

Henceforth, we obtain for the variance:

$$\mathbb{V}\mathrm{ar}[\mathrm{S}_{\cos}(\boldsymbol{g}, \boldsymbol{h})] = \frac{\sum_i \sigma_{\boldsymbol{g},i}^2(\sigma_{\boldsymbol{h},i}^2 + \mu_{\boldsymbol{h},i}^2) + \sigma_{\boldsymbol{h},i}^2 \mu_{\boldsymbol{g},i}^2}{\sum_i \mu_{\boldsymbol{g},i}^2 + \sigma_{\boldsymbol{g},i}^2 \sum_i \mu_{\boldsymbol{h},i}^2 + \sigma_{\boldsymbol{h},i}^2}. \tag{29}$$

To empirically assess the approximation quality, we compared the approximation to a kernel density estimate (KDE) over Monte Carlo samples. In particular, we generated $500$ samples for the image and text feature distributions for a given input. For the resulting samples, we then computed the respective cosine similarity for each pair and performed kernel density estimation with Gaussian kernel and lengthscale of $0.3$ on the similarity scores. We added increasing shifts to the distribution mean to evaluate the change in the approximation quality under varying cosine similarity values. Fig. 4 illustrates the approximation quality compared to a Monte Carlo simulation for image-text pairs with increasing distance between their feature projection means.

## D   Details on Support Set Selection

This section provides further details on the support set selection strategies used in this work.

### D.1   k-Nearest Selection

Active learning acquisition functions like Maximum Entropy Selection or BALD are often applied to the training set, lacking consideration of the target distribution and resulting in unrepresentative selections. To address this, we propose the following heuristic: we greedily acquire a maximally informative intermediate set $\mathcal{S}^* \subseteq \mathcal{X}_{\mathrm{test}}$ from the test set, followed by selecting training data points in the vicinity of the intermediate set $\mathcal{S}^*$. In case of deterministic embeddings one can use the cosine similarity or Euclidean distance for this purpose. However, as the embeddings are probabilistic in our setting, a point-wise comparison is not possible. Henceforth, we propose to compute the Wasserstein distance between the distributions of the embeddings of the test set and the training set, and select the training samples with minimal Wasserstein distance to the test set. For multivariate Gaussian distributions, the Wasserstein distance can be computed in closed form and is given as:

$$W_2^2\left(\mathcal{N}(\boldsymbol{\mu}_1, \boldsymbol{\Sigma}_1), \mathcal{N}(\boldsymbol{\mu}_2, \boldsymbol{\Sigma}_2)\right) = \|\boldsymbol{\mu}_1 - \boldsymbol{\mu}_2\|_2^2 + \mathrm{tr}\left(\boldsymbol{\Sigma}_1 + \boldsymbol{\Sigma}_2 - 2(\boldsymbol{\Sigma}_1^{1/2}\boldsymbol{\Sigma}_2\boldsymbol{\Sigma}_1^{1/2})^{1/2}\right) \tag{30}$$

where $\|\cdot\|_2$ denotes the Euclidean norm, $\mathrm{Tr}(\cdot)$ is the trace operator, and $\boldsymbol{\Sigma}^{1/2}$ is the matrix square root of $\boldsymbol{\Sigma}$. As computing the Wasserstein distance exactly is computationally and memory intensive, we approximate it by ignoring the correlation terms between the dimensions of the embeddings

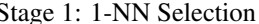

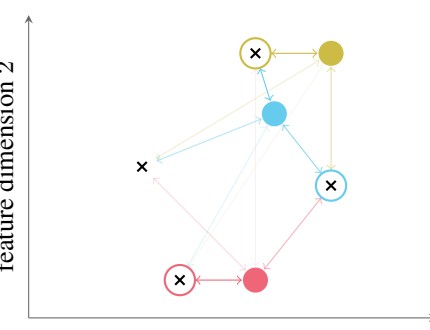

Stage 1: 1-NN Selection                          Stage 2: 2-NN Selection

Figure 5: Illustration of the nearest neighbour based support set selection for adaptive targeted selection. The circles ● show test data points with uncertainty scores depicted through their colours: high, medium, low. For each test datum we find the $k = 1$ nearest neighbour from the support set candidates ✗. If the $k = 1$ nearest neighbour is already selected, we increase $k$ for those with occupied neighbours and choose the second nearest neighbour, *i.e.*, $k = 2$. This recursion continues until every test datum has a selected support set candidate. The selected candidates are shown by coloured circles. Note that in case of the blue test datum, the closest support set candidate has already been chosen by the yellow and hence the second closes candidate is selected in the second stage.

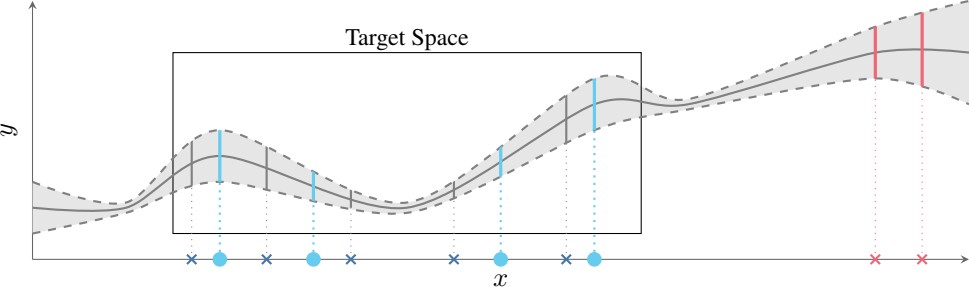

Figure 6: Illustration of targeted support set selection. We aim to select an informative support set that reduces the uncertainty over the predictions on the query set ●. Only focusing on the epistemic uncertainties would not lead to a good selection as we would select uninformative support set candidates ✗ with high epistemic uncertainty. Hence, we target the selection process.

resulting in the Wasserstein distance for univariate Gaussian distributions. We aim to explore more sophisticated approximations, *e.g.*, using the sliced Wasserstein distance [26], in future work. Based on this distance, we select the training samples closest to the test set in the joint embedding space, resulting in:

$$\mathcal{S} = \bigcup_{\boldsymbol{g}^* \in \mathcal{S}^*} \mathcal{N}_k(\boldsymbol{g}^*, \mathcal{X}_{\text{train}}), \tag{31}$$

with $\mathcal{N}_k(\boldsymbol{g}^*, \mathcal{X}_{\text{train}})$ denoting the set of $k$-nearest neighbours of $\boldsymbol{g}^*$ in the training set $\mathcal{X}_{\text{train}}$ according to the Wasserstein distance over the distributions of the normalized image embeddings. To ensure that we select $k$ distinct training samples for each test sample, we perform an iterative search in which we discard the already selected training samples and iteratively increase the search radius until $k$ distinct samples are found. This process is illustrated in Fig. 5.

### D.2 Acquisition Functions

**Naive Random**  For the *naïve random* acquisition function, we randomly sample $m$ data points from the train set $\mathcal{X}_{\text{train}}$ to form the support set $\mathcal{S}_{\text{ID}}$.

**Targeted Random**  For the *targeted random* acquisition function, we randomly sample $m$ data points from the test set $\mathcal{X}_{\text{test}}$ to form a intermediate support set $\mathcal{S}^*$. According to App. D.1, we then

select the nearest neighbours to $\mathcal{S}^*$ from the training set $\mathcal{X}_{\text{train}}$ based on the cosine similarity of the normalized image embeddings to form the support set $\mathcal{S}_{\text{t-ID}}$.

**Targeted Maximum Entropy** For the *entropy* acquisition function, we compute the predictive entropy $\mathcal{H}(y_i^* \mid \boldsymbol{x}_i^*)$ for each data point $\boldsymbol{x}_i^* \in \mathcal{X}_{\text{test}}$ and select the $m$ data points with the highest entropy. We use the predictive entropy on the MAP estimate of the model parameters to estimate the predictive entropy of the model:

$$\mathcal{H}\left(y \mid \boldsymbol{x}, \boldsymbol{\theta}_{\text{MAP}}\right) = -\sum_{c=1}^{C} p(y = c \mid \boldsymbol{x}, \boldsymbol{\theta}_{\text{MAP}}) \log p(y = c \mid \boldsymbol{x}, \boldsymbol{\theta}_{\text{MAP}}) \tag{32}$$

According to App. D.1, we then select the most similar data points from $\mathcal{X}_{\text{train}}$ to form the support set $\mathcal{S}_{\text{t-entropy}}$.

**BALD** We compute the BALD score [15] for each data point in $\mathcal{X}_{\text{train}}$ and select the $m$ data points with the highest score. The score is approximated using nested Monte Carlo sampling as in [15].

$$\text{BALD}(\boldsymbol{x}) = \mathbb{E}_{p(y|\boldsymbol{x})}\left[\mathcal{H}\left(p(\boldsymbol{\theta})\right) - \mathcal{H}\left(p(\boldsymbol{\theta} \mid \boldsymbol{x}, y)\right)\right] \tag{33}$$

$$= \mathbb{E}_{p(\boldsymbol{\theta}|\mathcal{D})}\left[\mathcal{H}\left(p(y \mid \boldsymbol{x}, \boldsymbol{\theta})\right) - \mathcal{H}\left(p(y \mid \boldsymbol{x}, \mathcal{D})\right)\right] \tag{34}$$

**Targeted BALD** We compute the BALD score (Eq. (34)) for each data point $\boldsymbol{x}_i^* \in \mathcal{X}_{\text{test}}$ and select the $m$ data points with the highest score. According to App. D.1, we then select the most similar data points from $\mathcal{X}_{\text{train}}$ to form the support set $\mathcal{S}_{\text{t-BALD}}$.

**EPIG** The Expected Predictive Information Gain (EPIG) score [3] calculates the expected mutual information between the model parameters and the predictive distribution resulting from the acquisition of a training data point. This method is specifically designed to target relevant information, eliminating the need for a k-nearest neighbor search typically used in other acquisition functions. The EPIG score is given by

$$\text{EPIG}(\boldsymbol{x}) = \mathbb{E}_{p_*(\boldsymbol{x}^*)p_\phi(y|\boldsymbol{x})}\left(\mathcal{H}\left(p_\phi(y^* \mid \boldsymbol{x}^*)\right) - \mathcal{H}\left(p_\phi(y^* \mid \boldsymbol{x}^*, x, y)\right)\right) \tag{35}$$

$$= \mathbb{E}_{p_*(\boldsymbol{x}^*)}\left[\text{D}_{\text{KL}}\left(p_\phi(y, y^* \mid \boldsymbol{x}, \boldsymbol{x}^*) \,\|\, p_\phi(y \mid \boldsymbol{x})p_\phi(y^* \mid \boldsymbol{x}^*)\right)\right] \tag{36}$$

$$= \mathbb{E}_{p_*(\boldsymbol{x}^*)}\left[\sum_{y \in \mathcal{Y}} \sum_{y^* \in \mathcal{Y}} p_\phi(y, y^* \mid \boldsymbol{x}, \boldsymbol{x}^*) \log \frac{p_\phi(y, y^* \mid \boldsymbol{x}, \boldsymbol{x}^*)}{p_\phi(y \mid \boldsymbol{x})p_\phi(y^* \mid \boldsymbol{x}^*)}\right] \tag{37}$$

and can be approximated using Monte Carlo sampling. For the EPIG selection we perform online updates to the model weights using the online Laplace as described in App. C.1.2.

# E Experiments

## E.1 Experimental Details

In our experiments we used the a pre-trained CLIP model [30] as the vision-language model with a ViT-Base and ViT-Huge backbone. We estimated the Hessians separately for the CLIP image and text encoders using the pre-training dataset Laion-400M [35]. For this estimation, we randomly sampled a subset of 3 million data points for the CLIP model with a ViT-Base backbone and 0.5 million data points for the CLIP model with a ViT-Huge backbone. The pre-training dataset was filtered to exclude NSFW content. For the Laplace approximation, we used the GGN approximation of the Hessian matrices as described in Sec. 2 and estimated the covariance matrices $\boldsymbol{A}$ and $\boldsymbol{B}$ for the image and text encoders. We use a grid search to find the Hessian scaling $\tau$ and learned the optimal prior precision by maximizing the marginal likelihood of the training data. The grid for the Hessian scale was set to $\tau \in \{0.3, 0.35, 0.4, 0.45, 0.5\}$ for the ViT-Base model and $\tau \in \{0.6, 0.65, 0.7, 0.75, 0.8\}$ for the ViT-Huge model.

For the *Office-Home* and *Flowers* data sets, we used the pre-defined splits provided by the original authors. For *EuroSAT*, we utilized the splits provided by [37]. For *ImageNet-R*, we divided the provided training set into a training and validation set with a validation ratio of $0.25$ and used the provided test set as is. Similarly, for the *Food* and *CIFAR-10/100* data sets, we split the training set into a training and validation set with a validation ratio of $0.2$ and used the provided test set without modifications.

Table 1: Data specifications for finetuning data sets with the number of classes $c$, training set size $n_{\text{train}}$, validation set size $n_{\text{val}}$, and test set size $n_{\text{test}}$.

| Dataset | $c$ | $n_{\text{train}}$ | $n_{\text{val}}$ | $n_{\text{test}}$ |
|---|---|---|---|---|
| Flowers [27] | 102 | 1020 | 1020 | 6100 |
| Food-101 [5] | 101 | 75750 | 15150 | 25250 |
| CIFAR-10/100 [21] | 10/100 | 50000 | 10000 | 10000 |
| ImageNet-R [13] | 200 | 22500 | 4500 | 7500 |
| ImageNet1k (subset classes) | 200 | 11168 | 2792 | 2298 |
| EuroSAT [12] | 10 | 13500 | 8100 | 5400 |
| Office-Home (clipart) [39] | 65 | 2793 | 699 | 873 |
| Office-Home (product) [39] | 65 | 2840 | 711 | 888 |
| Office-Home (real world) [39] | 65 | 2788 | 697 | 872 |

In our experiments, we compare the performance of the proposed EPIG acquisition function to various baseline acquisition functions: **Naive Random**, **Targeted Random**, **Targeted Maximum Entropy**, **Targeted BALD**, **EPIG**, and **Targeted EPIG**.

**Finetuning Settings**   For the finetuning, we trained we create support sets of size $m \in \{10, 25, 50, 75, 100, 150, 200, 500, 1000\}$ using the cross-entropy loss for 100 epochs. For evaluation, we report performance of best checkpoint according to validation loss.

**Data sets**   We experiment with the following data sets: Flowers102 [27], Food101 [5], CIFAR-10/100 [21], ImageNet-R [13], EuroSAT [12] and Office-Home [39]. Table 1 shows the data split sizes and number of classes for each dataset.

**Metrics**   We evaluate each method by measuring the class-weighted accuracy (ACC) on the test set that weights the accuracy based on the number of samples per class. Moreover, we use the negative log predictive density (NLPD) to assess the quality of the uncertainty estimates. We report the performance of each finetuned method at the epoch with the lowest validation loss.

## E.2   Additional Results

This section provides additional experimental results and ablations of the proposed method.

**Cross-domain Finetuning Results**   Fig. 9 show additional results for the cross-domain setting on the Office-Home data set for both the base and huge variants of the OpenCLIP model.

**Single-domain Finetuning Results**   Fig. 7 and Fig. 8 show the results for single-domain finetuning with support set selection using the huge and base variants of the OpenCLIP model, respectively. We also show the zero-shot performances from the pretrained CLIP models without any finetuning on the target task (Zero-shot). Note that we only show the performance for EPIG without targeted support set selection, as we noticed that EPIG performs competetively against the other selection methods in this single-domain finetuning setting.

We observe that the selection methods using the epistemic uncertainty (BALD and EPIG) perform better or on par with the Targeted Maximum Entropy across the different subset sizes and data sets. The accuracy and NLPD become better when increasing the subset sizes, and the huge model variant (Fig. 7) achieves higher accuracies and lower NLPD on all data sets compared to the base model variant (Fig. 8) due to its larger model capacity. On EuroSAT, the Random baselines perform on par with EPIG which possibly is due to that EuroSAT has a small number of classes that can be similar, *e.g.*, the classes Sea/Lake and River. These results demonstrate the benefits of using our proposed uncertainty estimates for support set selection.

## E.3   Covariance Analysis

In addition to the presented experiments, we performed an ablation on the sensitivity of the covariance to perturbations in the inputs. As shown in App. E.3, we observe that the covariance over the cosine similarities encodes meaningful information about the uncertainty of the model predictions under input perturbations.

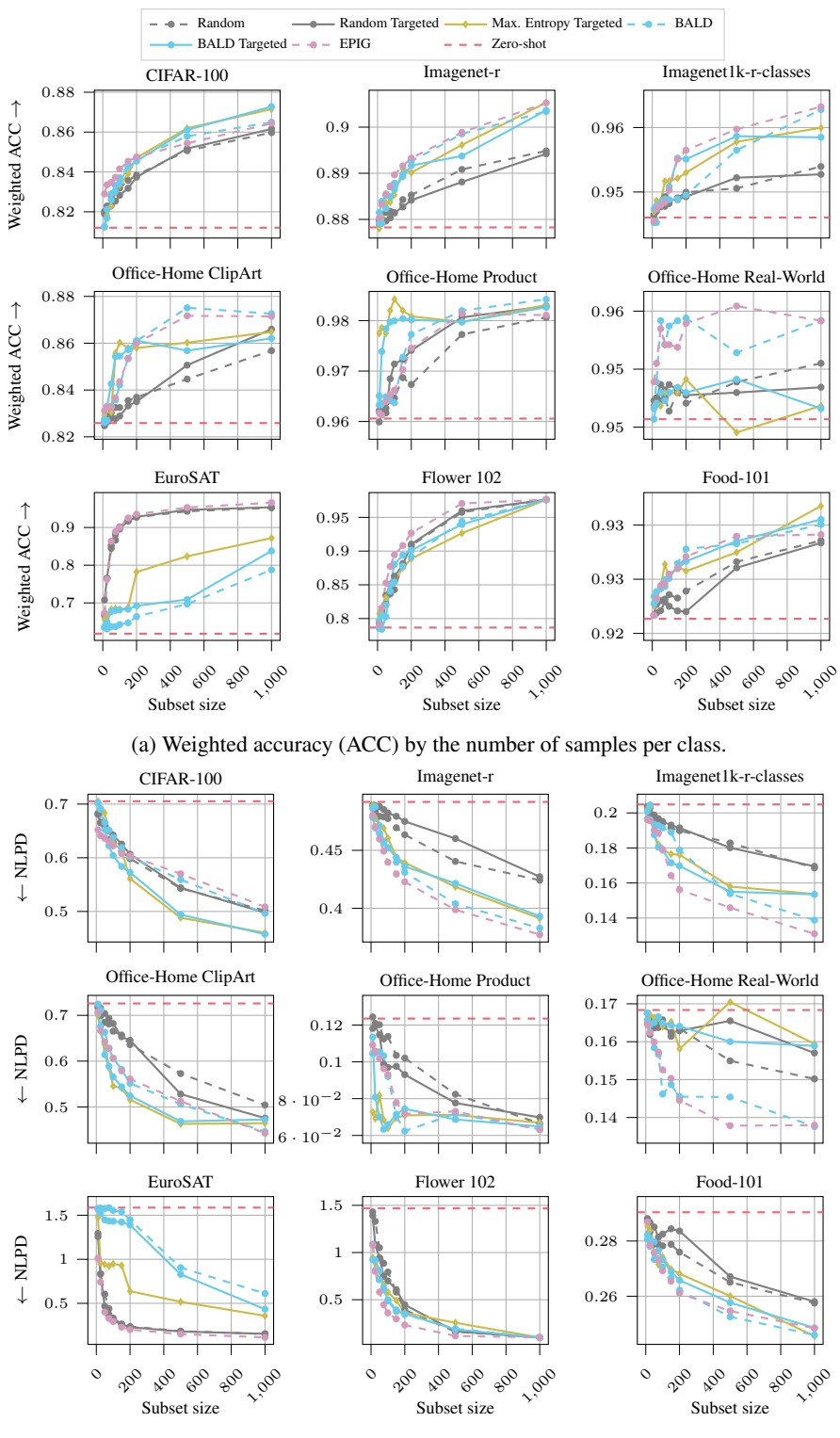

(a) Weighted accuracy (ACC) by the number of samples per class.

(b) Negative log-probability density (NLPD).

Figure 7: Accuracy and negative log-probability density (NLPD) over subset sizes of the support set across different data sets and subset selection methods using the OpenCLIP huge model variant. Results for random are averaged over 5 seeds.

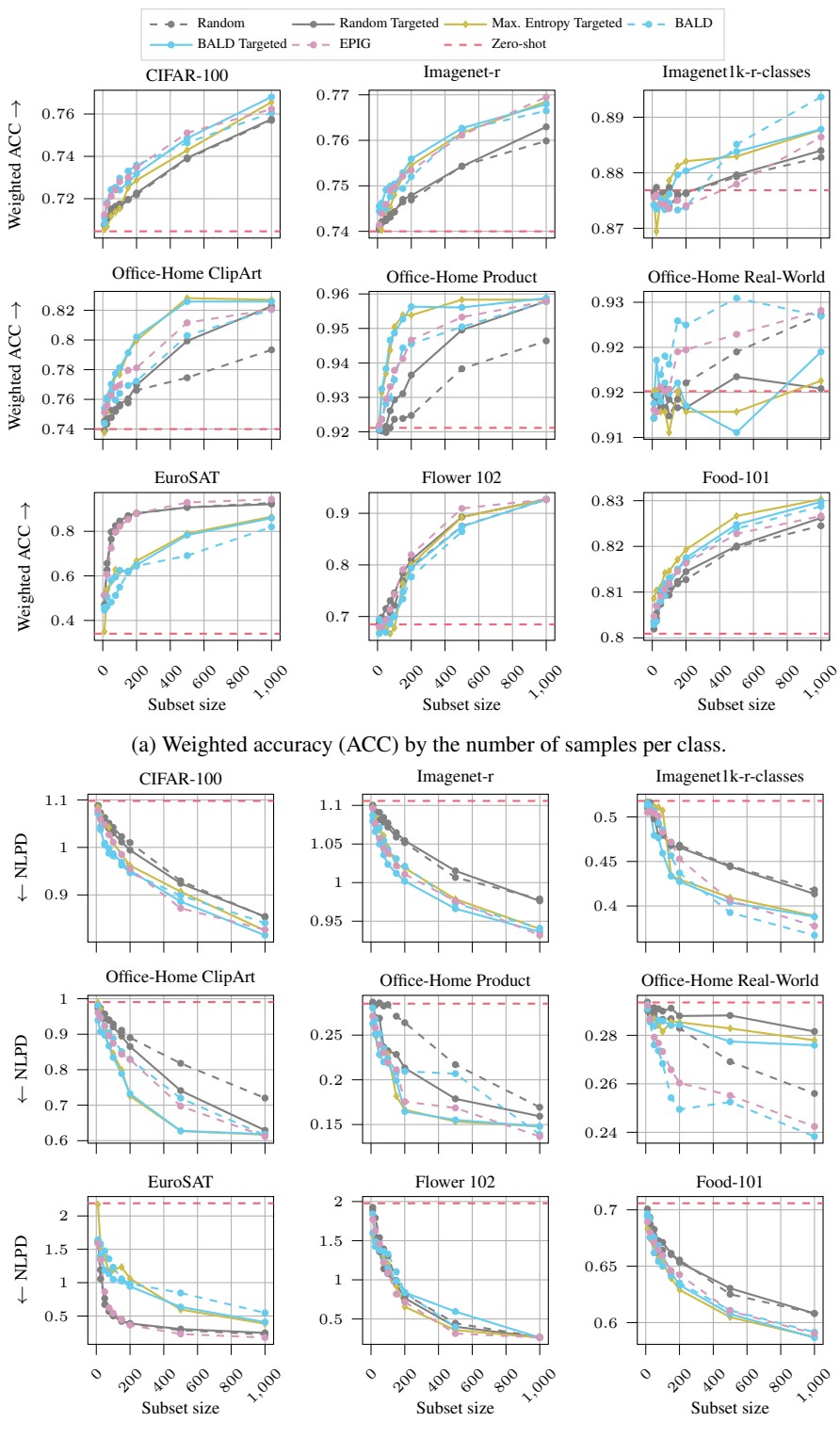

(a) Weighted accuracy (ACC) by the number of samples per class.

(b) Negative log-probability density (NLPD).

Figure 8: Accuracy and negative log-probability density (NLPD) over subset sizes of the support set across different data sets and subset selection methods using the OpenCLIP base model variant. Results for random are averaged over 5 seeds.

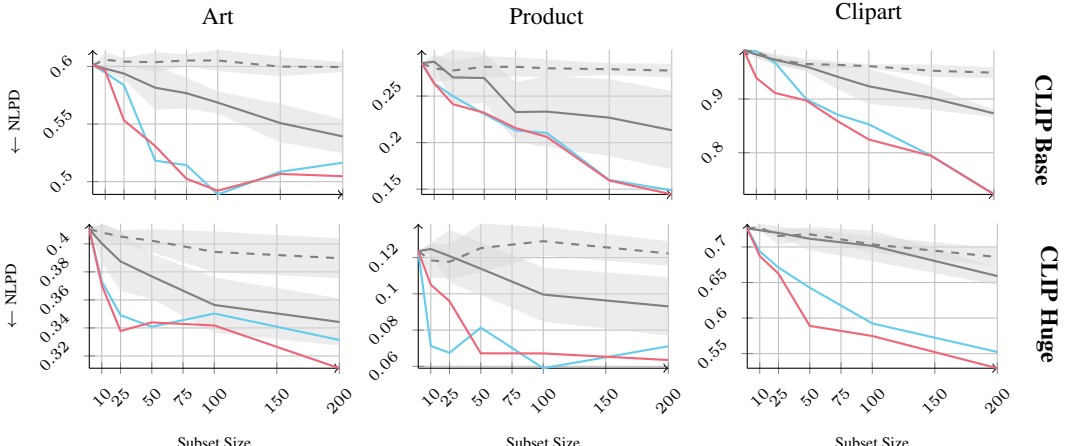

Figure 9: Results on the Office-Home data set with support set selection from all training domains. We depict the performance of the best performing acquisition function incorporating epistemic uncertainties (——), entropy based selection with targeted support set region (——), naïve random selection (- - -), and random selection with targeted support set candidates (——).

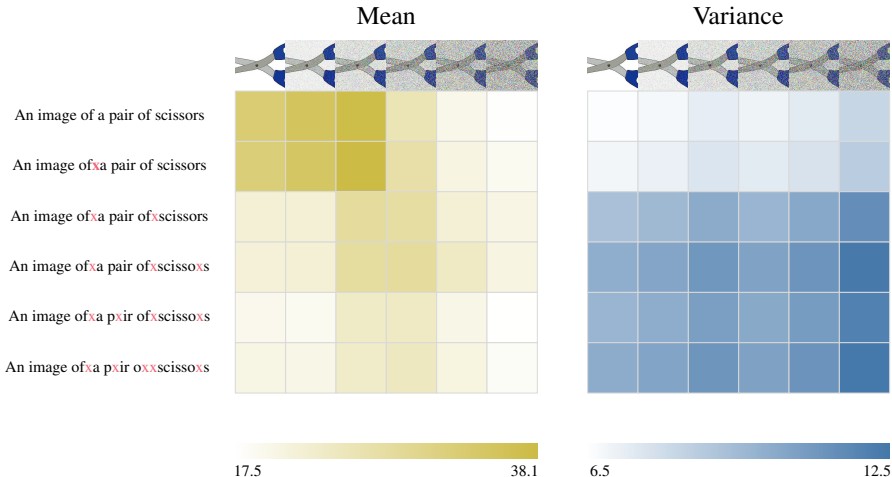

Figure 10: Illustration of the distribution over cosine similarities, depicting mean and variance, for varying image and text perturbations. We can observe that the mean cosine similarity decreases with increasing perturbation, while the variance increases, indicating that the distribution over cosine similarities captures model uncertainties in out-of-distribution settings.

