# OpenReview forum: "Probabilistic Active Few-Shot Learning in Vision-Language Models"
_NeurIPS.cc/2024/Workshop/BDU — NeurIPS BDU Workshop 2024 Poster_

### Official Review · Reviewer_pGjU · 2024-09-18
**This paper makes a good contribution to the area of active few-shot learning by proposing a novel method for uncertainty estimation in VLMs, which leads to improved model performance. The combination of Bayesian uncertainty estimation and targeted support set selection is a key strength, and the experimental results demonstrate the method’s effectiveness.**

**Rating:** 8
**Confidence:** 5

**Review:**

This paper is technically sound and demonstrates a deep understanding of active learning and uncertainty estimation in Vision-Language Models (VLMs). The authors provide a thorough theoretical foundation, leveraging Bayesian methods (specifically the Laplace approximation) to estimate uncertainty in a post-hoc manner. The mathematical rigor in the derivations of the Gaussian approximation over cosine similarities and the use of the Generalized Gauss-Newton (GGN) approximation demonstrate high quality and precision in the proposed methodology. The experimental evaluation is comprehensive, covering different data sets, models, and settings.
The post-hoc application of Bayesian uncertainty estimation methods, such as the Laplace approximation, in the context of Vision-Language Models (VLMs), without needing to retrain the models. While previous work on active learning and VLMs exists, this study introduces a new angle by combining active learning with probabilistic embeddings and exploring the role of uncertainty in support set selection. Furthermore, the use of Wasserstein distance-based selection of support set candidates adds novelty to the work. The approach of incorporating epistemic uncertainty into few-shot learning and targeting support set selection toward query regions represents a fresh and significant contribution to the field.
The paper’s approach of utilizing probabilistic embeddings and epistemic uncertainty could potentially make few-shot learning more efficient and reliable, which is crucial for real-world applications where labeled data is scarce.

---

### Official Review · Reviewer_FUYj · 2024-09-22
**Overall, it is a nice approach with decent empirical results**

**Rating:** 7
**Confidence:** 4

**Review:**

Pros:
- Using Bayesian modeling (even crude approximations of it) is highly suitable for active learning. Combining it with tasks related to contemporary NNs is a great direction.
- Choosing the Laplace approximation (with the appropriate relaxations based on known solutions in the literature) is suitable for the case presented in the paper and makes sense.
- Because of the page limit, the main paper may be a little hard to follow for researchers from the general DL community (i.e., ones who are not familiar with Bayesian modeling), but the appendix covers the required background well and presents the derivations clearly.
- The experiments are extensive.
- For the most part, the paper is written clearly.

Cons:
- The main limitation of the paper are the empirical results. If I understand the results correctly, there is a slight advantage for the proposed approach over the entropy-based baseline. Also, I didn't see in the paper a comparison to other methods in the literature (possibly non-Bayesian) in case there are any.
- Minor:
  - Reference in line 69 should be to Appendix B1
  - ln line 335 a reference is missing
  - $\hat{H}$ in line 341 is not defined anywhere.

Questions:
- I wonder if other types of posterior approximations were used, such as a diagonal Laplace approximation over the entire network.
- Why are you showing only class-weighted accuracy? Put differently, I believe it will be interesting to see the general (unweighted) accuracy as well.

---

### Decision · Program_Chairs · 2024-10-09

**Decision:**

Accept (Poster)

**Comment:**

Reviews are unanimously positive. I concur.